# Oxidation Catalysis of Au Nano-Particles Immobilized on Titanium(IV)- and Alkylthiol-Functionalized SBA-15 Type Mesoporous Silicate Supports

Tomoki Haketa, Toshiaki Nozawa, Jun Nakazawa ⬤, Masaya Okamura and Shiro Hikichi *

Department of Material and Life Chemistry, Faculty of Engineering, Kanagawa University, 3-27-1 Rokkakubashi, Kanagawa-ku, Yokohama 221-8686, Japan
* Correspondence: hikichi@kanagawa-u.ac.jp; Tel.: +81-45-481-5661

**Abstract:** Novel Au nano-particle catalysts immobilized on both titanium(IV)- and alkylthiol-functionalized SBA-15 type ordered mesoporous silicate supports were developed. The bi-functionalized SBA-15 type support could be synthesized by a one-pot method. To the synthesized supports, Au was immobilized by the reaction of the alkylthiol groups on the supports with $AuCl_4{}^-$, following reduction with $NaBH_4$. The immobilized amount and the formed structures and the electronic property of the Au species depended on the loading of alkylthiol. The moderate size (2–3 nm) nano particulate Au sites formed on Ti(0.5)-SBA$^{SH}$(0.5) were negatively charged. The aerobic alcohol oxidation activity of the catalysts depended on the loading of alkylthiol and the structure of the Au nano-particles. The non-thiol-functionalized catalyst (Au/Ti(0.5)-SBA$^{SH}$(0)) composed of the large (5–30 nm) and the higher thiol-loaded catalyst (Au/Ti(0.5)-SBA$^{SH}$(8)) composed of the small cationic Au species were almost inactive. The most active catalyst was Au/Ti(0.5)-SBA$^{SH}$(0.5) composed of the electron-rich Au nano-particles formed by the electron donation from the highly dispersed thiol groups. Styrene oxidation activity in the presence of 1-phenylethanol with $O_2$ depended on the loadings of titanium(IV) on the Au/Ti($x$)-SBA$^{SH}$(0.5). The titanium(IV) sites trapped the $H_2O_2$ generated through the alcohol oxidation reaction, and also contributed to the alkene oxidation by activating the trapped $H_2O_2$.

**Keywords:** Au nano-particles; aerobic oxidation; functionalized mesoporous silicate

## 1. Introduction

The catalysis of nano-scale Au particles has attracted much attention. In particular, the oxidation catalysis of Au nano-particles is interesting because of their applicability toward various oxidation reactions including dehydrogenative oxidation of alcohol and oxygenation of hydrocarbons [1–10]. Catalytic performances of Au particles are affected by the size of the particles. To control the particle sizes and prevent aggregating the particles, interactions between Au and soft donors have been utilized [11–16]. Particularly, organic thiol-modified silicates have been employed as precursors of solid supports of Au nano-particle catalysts [15,16]. During the immobilization process, Au(III) precursors are reduced to Au(I) species by thiols and stable Au–thiolate complexes are formed. Following reduction in the Au–thiolate compounds yield Au(0) nano-particles. In the Au catalysts prepared by these procedures, loadings of the thiols are relatively high (~ 1 mmol g$^{-1}$), and these thiol functionalities are removed by calcination [15,16]. The remaining thiols are believed to have negative effects on the catalysis due to changing the electronic properties of the Au particle and preventing the access of substrates to active sites. However, a correlation between the thiol loadings and properties of the supported Au nano-particle catalysts has not been investigated systematically.

To improve the activity toward hydrocarbon oxidation, Au nano-particle catalysts are combined with other active species, such as titanium(IV) sites of titanosilicates [17,18]. Sev-

---

eral titanium(IV) compounds are known to catalyze the alkene epoxidation with hydrogen peroxide in homogeneous and heterogeneous system [19–25]. The heterogeneous titanosilicate compounds have been reported to catalyze not only the epoxidation of alkenes, but also the oxygenation of alkanes and aromatic rings using hydrogen peroxide as an oxidant [24–28]. During these catalytic reactions based on titanium(IV), peroxido complexes of titanium(IV) might form as the reaction intermediate. It is known that aerobic alcohol oxidation is catalyzed by Au nano-particle catalysts, and during the alcohol oxidation, $H_2O_2$ would be formed and the resulting generated $H_2O_2$ could be utilized as an oxidant in situ. Therefore, a combination of Au nano-particle catalysts and other species which activate $H_2O_2$ would become an aerobic oxidation catalyst applicable to various substrates including alkenes and alkanes. In this work, we have designed novel Au nano-particle catalysts immobilized on both titanium(IV)- and alkylthiol-functionalized SBA-15 type mesoporous silicates.

## 2. Results

### 2.1. Preparation and Characterization of the Catalysts

2.1.1. Preparation and Characterization of Bi-Functionalized SBA-15 Type Supports

The titanium(IV)- and alkylthiol-functionalized SBA-15 type supports Ti($x$)-SBA$^{SH}$($y$), of which $x$ and $y$ denote the molar ratio of the titanium and alkylthiol sources, respectively, were synthesized by the one-pot condensation of alkoxide precursors in the presence of polymer micelle templates. Molar ratio of precursors Si(OEt)$_4$: Ti(O$i$Pr)$_4$: Si(C$_3$H$_6$SH)(OMe)$_3$ were controlled as $100 - (x + y)$: $x$: $y$ (Scheme 1).

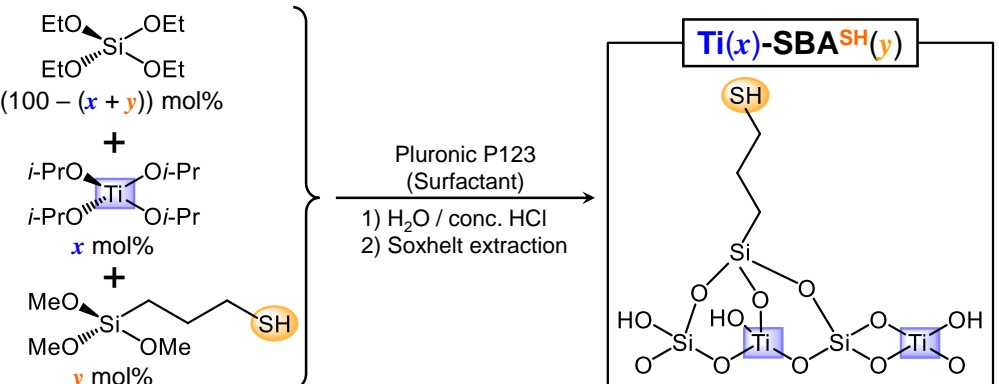

**Scheme 1.** Preparation of the bi-functionalized supports Ti($x$)-SBA$^{SH}$($y$).

The contents of titanium(IV) and alkylthiol, and the physicochemical properties of Ti(0.5)-SBA$^{SH}$($y$), prepared by using 0.5 mol% of Ti(O$i$Pr)$_4$ and 0–10 mol% of Si(C$_3$H$_6$SH) (OMe)$_3$ are shown in Table 1. The nitrogen adsorption/desorption isotherms of the obtained supports showed type IV hysteresis, confirming the presence of mesopores (Figure S1). The surface area, pore size distribution, and pore volume derived from the isotherms were also consistent with the characteristics of SBA-15 type mesoporous silica support except for Ti(0.5)-SBA$^{SH}$(10), of which the surface area and pore volume were decreased significantly (Figure 1a, see also XRD analysis data shown as Figure S2 in Supplementary Materials). The loading amounts of alkylthiol and titanium(IV) were quantified by analysis of the solution of the supports obtained by alkaline treatment with the method on $^1$H NMR for alkylthiol and UV-vis spectroscopy for titanium(IV). The loadings of alkylthiol were proportional to the amount of Si(C$_3$H$_6$SH)(OMe)$_3$ applied on the preparation when $y$ was less than 8. The loading amount of alkylthiol on Ti(0.5)-SBA$^{SH}$(10) was, however, decreased due to partial destruction of the SBA-15 type mesoporous channel structure. Furthermore, the anchored alkylthiol did not affect the loading of titanium(IV) as evidenced by the almost constant amounts of titanium involved in Ti(0.5)-SBA$^{SH}$($y$).

**Table 1.** The contents of functionalities and the physicochemical properties on Ti(*x*)-SBA$^{SH}$(*y*).

| Applied Ratio/mol% | | Contents/mmol g$^{-1}$ | | Surface Area/m$^2$ g$^{-1}$ | Average Pore Diameter/nm | Total Pore Volume /cm$^3$ g$^{-1}$ |
|---|---|---|---|---|---|---|
| *x* | *y* | Ti | Alkylthiol | | | |
| | 0 | 0.064 | — | 847.3 | 8.57 | 0.82 |
| | 0.25 | 0.062 | 0.035 | 1013.1 | 10.09 | 0.95 |
| | 0.5 * | 0.066 * | 0.064 * | 971.4 * | 9.58 * | 1.04 * |
| | 0.75 | 0.063 | 0.101 | 1058.3 | 8.96 | 0.97 |
| 0.5 | 1 | 0.068 | 0.121 | 948.1 | 9.54 | 1.15 |
| | 4 | 0.066 | 0.486 | 710.2 | 9.46 | 0.71 |
| | 8 | 0.056 | 0.900 | 726.3 | 8.14 | 0.99 |
| | 10 | 0.058 | 0.761 | 415.8 | 6.41 | 0.43 |
| 0 | | — | 0.062 | 897.5 | 10.05 | 0.91 |
| 0.5 * | | 0.066 * | 0.064 * | 971.4 * | 9.58 * | 1.04 * |
| 1 | 0.5 | 0.152 | 0.062 | 795.2 | 9.64 | 0.79 |
| 2 | | 0.309 | 0.063 | 819.2 | 7.48 | 0.85 |
| 3 | | Not measured | | 732.0 | Not calculated | 0.56 |

\* Same sample of Ti(0.5)-SBA$^{SH}$(0.5).

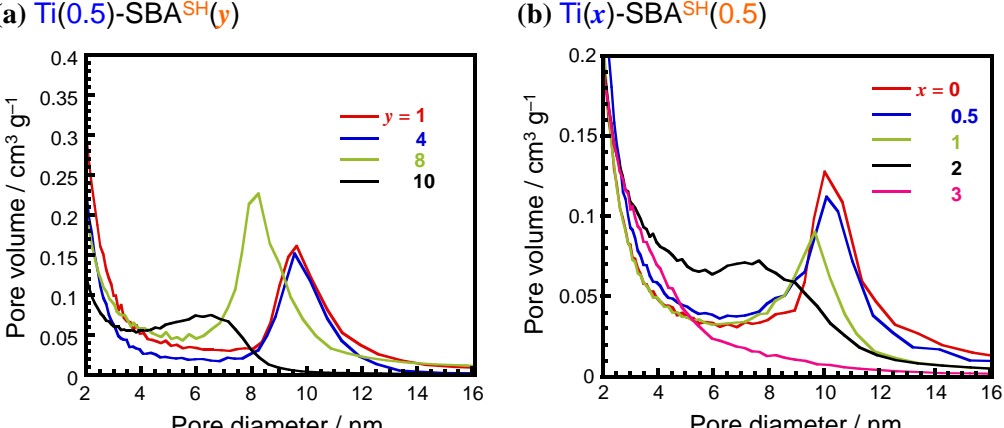

**(a)** Ti(0.5)-SBA$^{SH}$(*y*)    **(b)** Ti(*x*)-SBA$^{SH}$(0.5)

**Figure 1.** The distribution of the pore diameters of the bi-functionalized supports. (**a**) Ti(0.5)-SBA$^{SH}$(*y*); (**b**) Ti(*x*)-SBA$^{SH}$(0.5).

Titanium-loading controlled supports, Ti(*x*)-SBA$^{SH}$(0.5) were prepared with the applied amount of alkyl thiols fixed at 0.5 mol% and with varying applied amounts (*x*) of Ti(O*i*Pr)$_4$ from 0 to 3 mol% on the preparation (Figures S2 and S4). The pore diameter distributions indicate that the loadings of titanium(IV) over 2.0 mol% lead to the destruction of the SBA-15 type structure as shown in Figure 1b because Ti(IV) favors a six-coordinate structure, and that leads to the formation of TiO$_2$ crystalline domains as reported previously [29,30]. The characteristics of Ti(*x*)-SBA$^{SH}$(0.5) are summarized in Table 1. The loading amounts of titanium(IV) were proportional to the applied ratio of Ti(O*i*Pr)$_4$ on the preparation whereas the incorporated amounts of alkylthiol were almost constant.

### 2.1.2. Preparation and Characterization of Au-Immobilized Catalysts

The reaction of the prepared thiol-functionalized supports with yellow-colored EtOH solution of NaAuCl$_4$ (1.0 wt% based on the weight of the support was applied) yielded ionic Au-immobilized precursors. After filtration and washing, the color of the resulting ionic Au species-anchored supports was white. The observed color-changing behavior suggests that the thiols perform as a reducing agent (Au(III) + 2RSH → Au(I) + RSSR), and an anchor to form Au(I)-S(thiolate or disulfide) complexes.

Reduction in the Au(I)-anchored supports by NaBH$_4$ in EtOH yielded the catalysts Au/Ti(*x*)-SBA$^{SH}$(*y*). The color of the catalysts varied from reddish brown to pale brown on the thiol-functionalized supports. The non-thiol-functionalized support catalyst Au/Ti(0.5)-

SBA$^{SH}$(0)*, which was obtained by direct deposition of Au particles generated in situ through the reduction in the Au(III) ions by NaBH$_4$, colored purple. These colorations (shown in Figure 2c) are due to the surface plasmon resonance on the gold nano-particles, and the color difference is attributed to the difference in particle size.

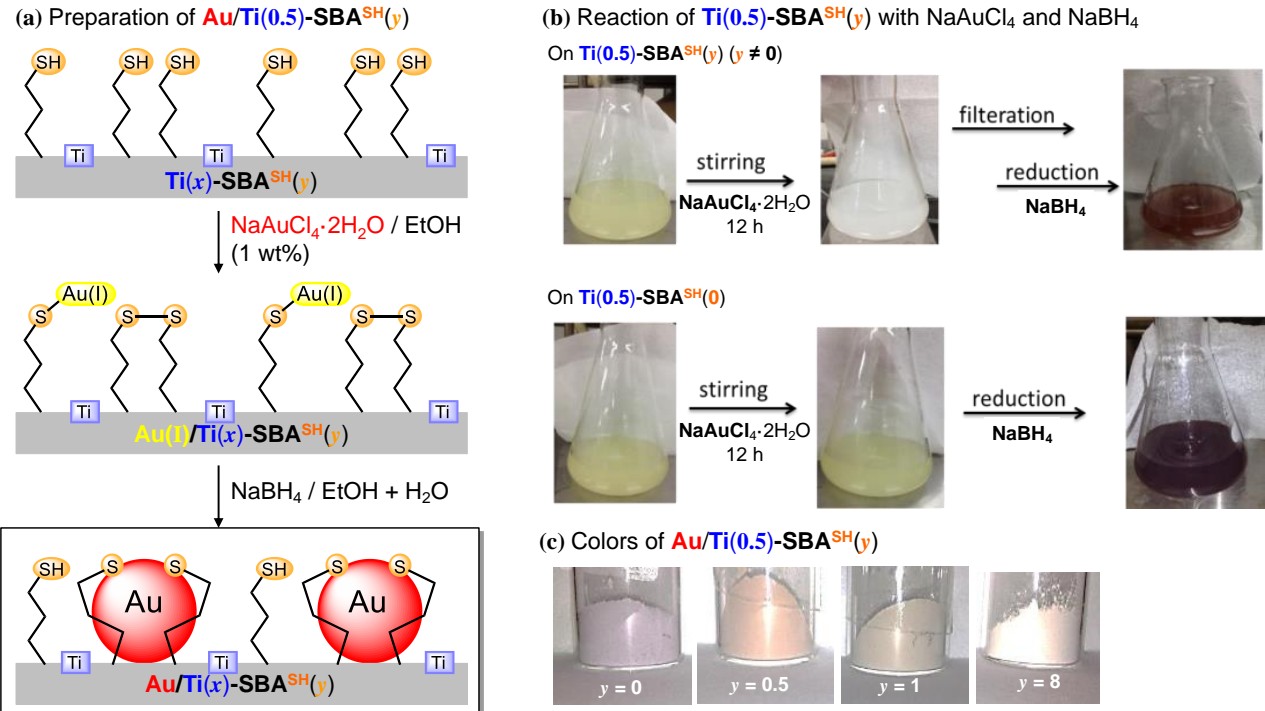

**Figure 2.** Preparation of Au-immobilized catalysts Au/Ti($x$)-SBA$^{SH}$($y$): (**a**) A synthetic procedure. (**b**) Reaction mixture of the supports with NaAuCl$_4$·2H$_2$O and reduction by NaBH$_4$. (**c**) Colors of the obtained Au-immobilized catalysts.

TEM and STEM images of Au/Ti(0.5)-SBA$^{SH}$($y$) revealed the correlation between the loading amount of alkylthiol and the size of formed Au nano-particles (Figure 3). On the lower thiol loading support ($y = 0.5$), the particle sizes of Au were 2–5 nm (Figure S5 in Supplementary Materials). The Au nano-particle sizes were decreased with the increased loading amount of thiol on the supports. On the higher alkylthiol loading support ($y = 8.0$), no Au particles larger than 1 nm in diameter could be observed. On the non-thiol-functionalized support ($y = 0$), the formed Au particles were larger than 5 nm.

The structures of the Au sites of the catalysts and their precursors (non-reduced Au ions immobilized) were analyzed by EXAFS (Figure 4). On the precursors, Au–S bonds were observed. On the lower thiol functionalized support ($y = 0.5$), the reduction with NaBH$_4$ led to reducing the Au–S bonds, whereas an increase in the Au–Au bonds was due to the formation of the Au nano-particles. In contrast, many Au–S bonds remained on the higher thiol-functionalized support ($y = 8$) even after the treatment with NaBH$_4$. A small extent of Au–Au bonds appeared, and, therefore, strong Au–S interaction would prevent the forming of the Au clusters. Electronic states of the resulting Au species were also varied according to the amounts of the alkylthiol loadings $y$. On the support with $y = 0.5$, the binding energy of XPS of the formed Au particles was slightly shifted to lower compared to that of the metallic Au formed on the non-thiol-functionalized support ($y = 0$). In contrast, increasing the thiol functionalities ($y = 8$) yielded cationic Au with higher binding energy as found in Figure 5.

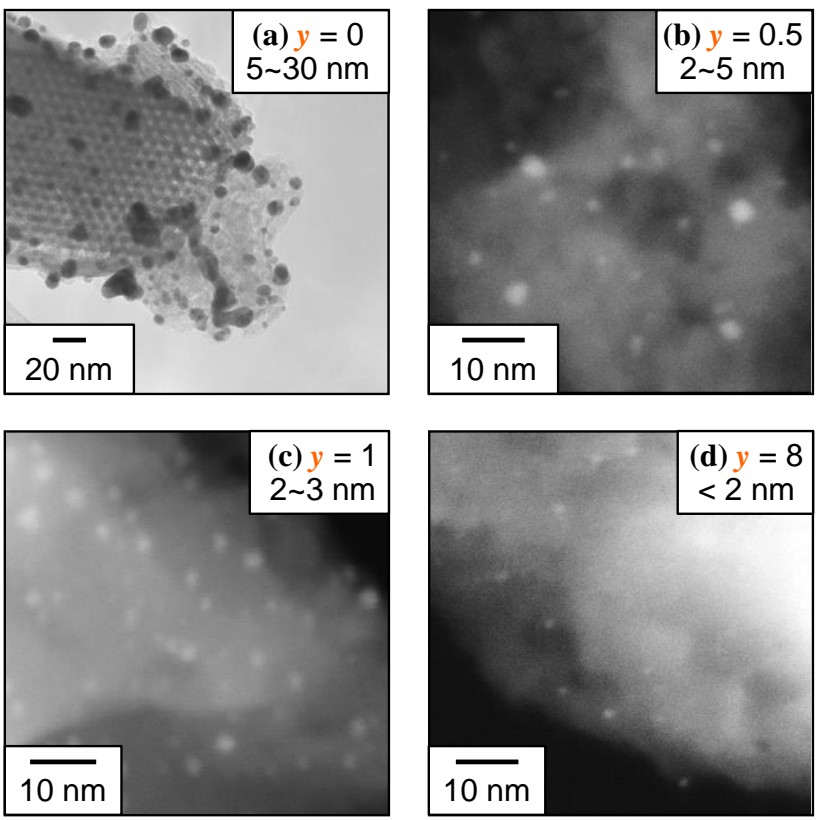

**Figure 3.** TEM (**a**) and STEM (**b**–**d**) images of Au/Ti(0.5)-SBA$^{SH}$(*y*) with *y* = 0, 0.5, 1, and 8.

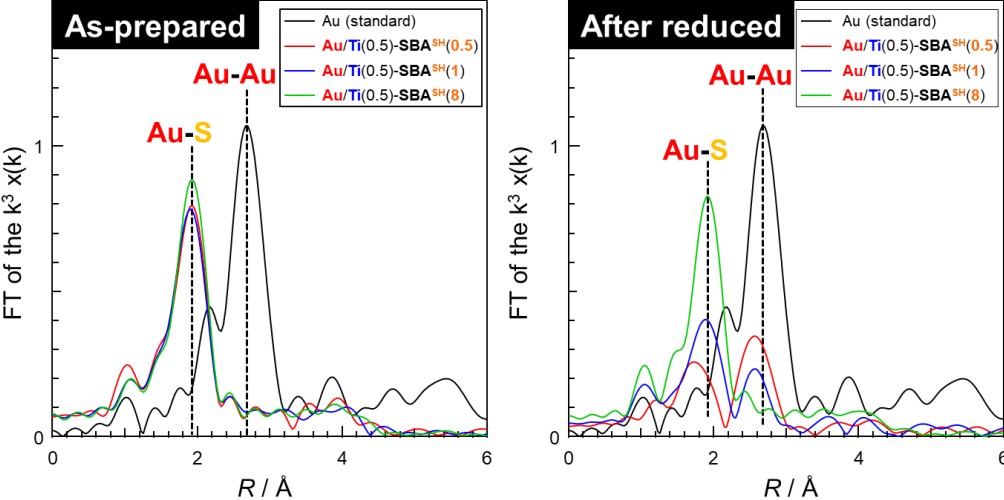

**Figure 4.** XAFS of Au/Ti(0.5)-SBA$^{SH}$(*y*) with *y* = 0, 0.5, 1, and 8.

The immobilized amount of Au also depended on the amount of alkylthiol on the supports. For the non-thiol-functionalized support (*y* = 0), a few Au species were immobilized on the solid filtered, and washed from the suspension of Ti(0.5)-SBA$^{SH}$(0) and NaAuCl$_4$ in EtOH. For Au/Ti(0.5)-SBA$^{SH}$(*y*) with 0 < *y* ≤ 4, the amount of the immobilized Au increased as increasing the alkylthiol content but was almost the same for the supports with *y* = 4 and *y* = 8 (Figure 6 and Table 2), respectively. On the titanium loadings varied catalysts Au/Ti(*x*)-SBA$^{SH}$(0.5) with 0 < *x* ≤ 2, the loading amount of Au was almost constant as shown in Table 2. The color of these catalysts was the same. Therefore, the loadings of titanium(IV) did not affect the immobilized amount and size of the Au nano-particles.

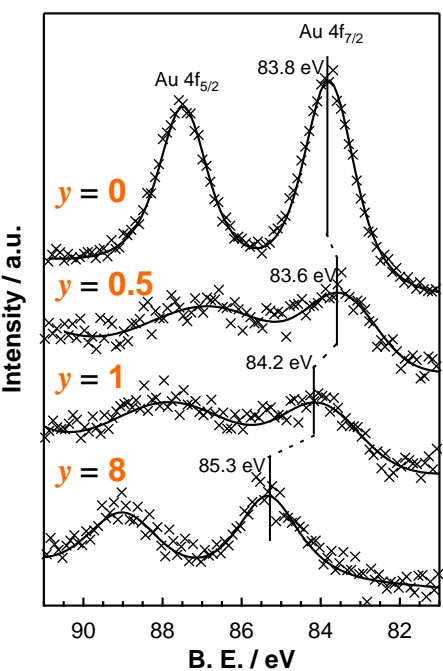

**Figure 5.** XPS of Au/Ti(0.5)-SBA$^{SH}$($y$) with $y$ = 0, 0.5, 1, and 8.

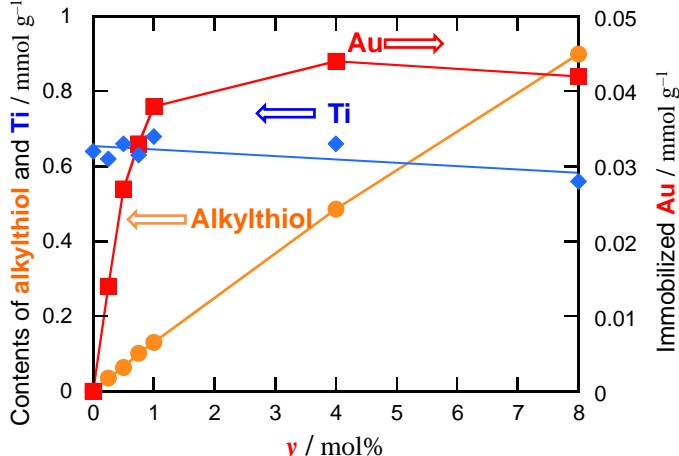

**Figure 6.** Correlation between the immobilized amount of Au and the contents of alkylthiol and titanium(IV) on Au/Ti(0.5)-SBA$^{SH}$($y$).

**Table 2.** Immobilized amounts of Au on Au/Ti($x$)-SBA$^{SH}$($y$).

| Catalyst | Alkylthiol/mmol g$^{-1}$ | Immobilized Au/mmol g$^{-1}$ |
|---|---|---|
| Au/Ti(0.5)-SBA$^{SH}$(0) | — | <0.001 |
| Au/Ti(0.5)-SBA$^{SH}$(0) [1] | — | 0.049 |
| Au/Ti(0.5)-SBA$^{SH}$(0.25) | 0.035 | 0.014 |
| Au/Ti(0.5)-SBA$^{SH}$(0.5) | 0.064 | 0.027 |
| Au/Ti(0.5)-SBA$^{SH}$(0.75) | 0.101 | 0.033 |
| Au/Ti(0.5)-SBA$^{SH}$(1) | 0.121 | 0.038 |
| Au/Ti(0.5)-SBA$^{SH}$(4) | 0.486 | 0.044 |
| Au/Ti(0.5)-SBA$^{SH}$(8) | 0.900 | 0.042 |
| Au/Ti(0)-SBA$^{SH}$(0.5) | 0.062 | 0.028 |
| Au/Ti(1)-SBA$^{SH}$(0.5) | 0.062 | 0.030 |
| Au/Ti(2)-SBA$^{SH}$(0.5) | 0.063 | 0.029 |

[1] Prepared by the deposition of in situ generated Au(0) particles.

### 2.1.3. Characterization of the Calcined Catalyst

TG analysis of the catalysts with $y$ = 1 and 8, respectively, indicated that the weight loss of the catalyst occurred over 423 K due to the combustion of alkylthiol. As described above, the particle size and immobilized amount of Au depended on the loading amount of alkylthiol. In addition, the EXAFS of Au/Ti(0.5)-SBA$^{SH}$(8) showed the Au–S bond. These observations suggest that alkylthiol interacts with the Au particles formed on the support. To remove alkylthiol, the calcination of the catalyst was examined. The higher alkylthiol loaded Au/Ti(0.5)-SBA$^{SH}$(8) was calcined at 673 K for two hours under air. The resulting calcined Au/Ti(0.5)-SBA$^{SH}$(8) was characterized by TEM and XPS. Formation of the nano Au particles over 2 nm diameter was observed on the TEM image, as shown in Figure 7a. The XPS data also indicated that the cationic Au species converted to the metallic one by calcination (Figure 7b).

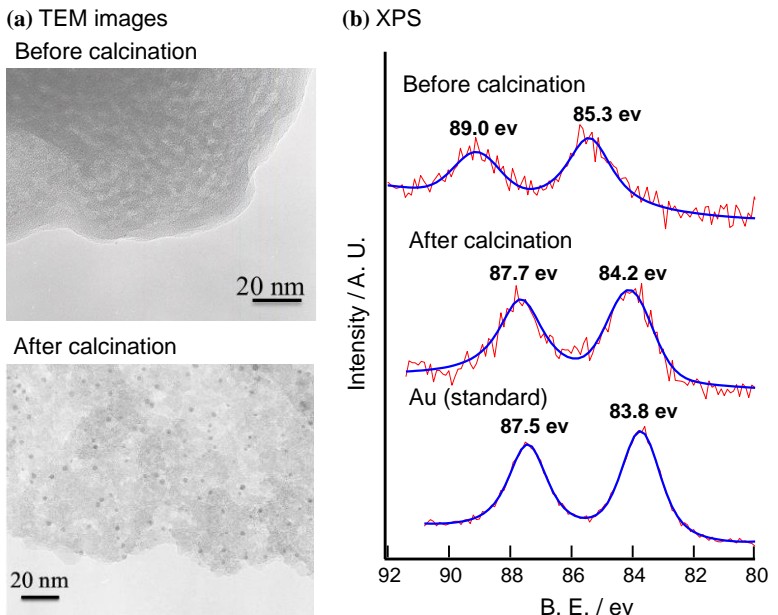

**Figure 7.** Comparison between the before and after calcined Au/Ti(0.5)-SBA$^{SH}$(8): (**a**) TEM images, (**b**) XPS data. The calcination was done at 673 K for 2 h.

### 2.2. Catalysis

#### 2.2.1. Aerobic Alcohol Oxidation Activity

Catalysis of the Au/Ti(0.5)-SBA$^{SH}$($y$) was assessed by oxidation of 1-phenylethanol with O$_2$ at 333 K (Figure 8). In our system, any base additives were not required. The non-thiol-functionalized ($y$ = 0) and the higher thiol-loaded ($y$ = 8) catalysts were almost inactive. The catalytic efficiency based on the immobilized amounts of Au atoms (=TON of Au) depended on the amounts of the thiol functionalities as $y$ = 0.25 ~ 0.5 > 0.75 > 1.0 >> 8.0 = 0 (Figure 8c).

The calcination of the Au/Ti(0.5)-SBA$^{SH}$($y$) ($y$ = 0.5, 1, and 8) at 573 or 673 K, respectively, for 2 h under air resulted in changing the catalytic performance. The order of the alcohol oxidizing activity of the catalysts calcined at 573 K was $y$ = 1 > 0.5 > 8. On the catalyst with $y$ = 0.5, the calcination led to a decrease in the activity. In contrast, the calcination resulted in increasing the activity on the catalyst with $y$ = 1, although the improved activity was still lower than the activity of the non-calcined catalyst with $y$ = 0.5. Notably, the calcined catalyst with $y$ = 8 showed acid catalysis rather than oxidation causing dehydrative condensation (giving compound **2**) and dehydration (giving compound **3**), as shown in Scheme 2 and Table 3. The origin of the acid catalysis might be the sulfate which was formed by the partial oxidation of thiol. Elongation of the calcination time resulted in increasing the alcohol oxidation activity with decreasing the acidic catalytic activity. Finally,

washing the 2 h calcined catalyst with $H_2O$ showed the highest activity in the calcined catalysts, although that was still lower than the non-calcinated catalyst with $y = 0.5$.

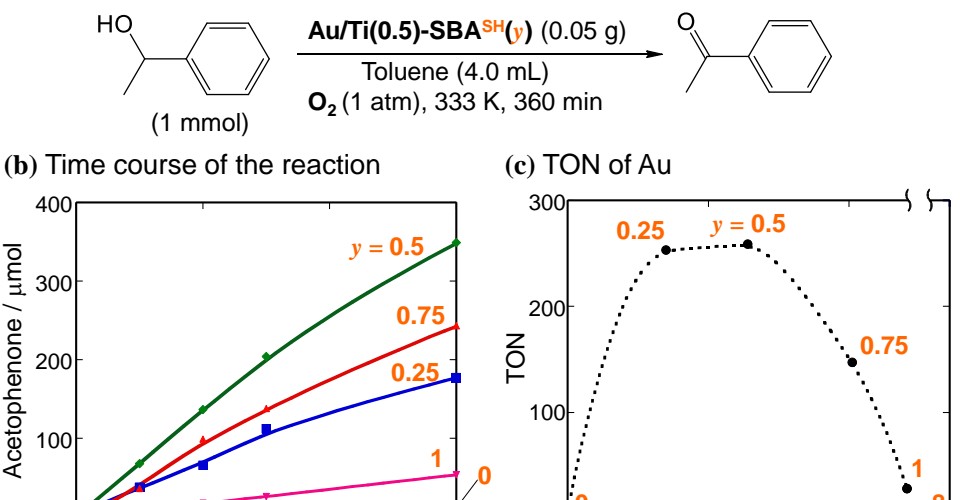

**Figure 8.** Aerobic oxidation of 1-phenylethanol catalyzed by Au/Ti(0.5)-SBA$^{SH}$($y$): (**a**) Reaction condition; (**b**) time course of the reaction; (**c**) content of alkylthiol versus TON of Au.

**Scheme 2.** Reaction of 1-phenylethanol catalyzed by the calcined Au/Ti(0.5)-SBA$^{SH}$($y$).

**Table 3.** Conversion of 1-phenylethanol mediated by the calcined catalysts under $O_2$.

| Catalyst | Calcination Condition | | Amount of Products/µmol | | TON of Au for 1 |
| --- | --- | --- | --- | --- | --- |
| | Temp./K | Time/h | 1 | 2 + 3 | |
| Au/Ti(0.5)-SBA$^{SH}$(0.5) | — | — | 349.1 | 0 | 258.6 |
| | 573 | 2 | 99.7 | trace | 73.9 |
| Au/Ti(0.5)-SBA$^{SH}$(1) | — | — | 53.6 | 0 | 29.8 |
| | 573 | 2 | 135.1 | trace | 75.1 |
| | 673 | 2 | 114.7 | trace | 63.7 |
| Au/Ti(0.5)-SBA$^{SH}$(8) | — | — | 0.2 | 0 | 0.1< |
| | 573 | 2 | 0.8 | 66.9 | 0.4 |
| | 673 | 2 | 4.7 | 166.1 | 2.3 |
| | 673 | 4 | 8.3 | 110.3 | 4.0 |
| | 673 | 8 | 26.7 | trace | 12.8 |
| | 673 | 12 | 0 | 0 | 0 |
| | 673 * | 2 * | 160.6 | 0 | 77.2 |

* This catalyst was washed with $H_2O$ after the calcination.

### 2.2.2. Co-Oxidation of Alcohol and Alkene

Then we investigated the role of the titanium(IV) sites by the assay of the styrene oxidation activity in the presence of 1-phenylethanol with $O_2$ (Scheme 3 and Table 4). In this reaction system, 1-phenylethanol might be a sacrificial reducing reagent and a hydrogen source for $H_2O_2$ generation. The yields of the styrene-oxidized products were correlated with the loading amounts of titanium(IV) (i.e., value of $x$), although the major product derived from styrene was benzaldehyde, and the yields of styrene oxide were quite low on each catalyst. When the non-Ti(IV)-loaded catalyst was used, the oxidation of styrene occurred [31]. The styrene oxidation activity was improved on the titanium(IV)-containing catalysts: The most reactive one was $x = 2.0$ in a series of Au/Ti($x$)-SBA^SH(0.5), although the major product was not epoxide **4**, but aldehyde **5**. In the absence of 1-phenylethanol, the yields of styrene oxidized products were low even on the catalyst with $x = 1.0$. Acetophenone was not formed in the absence of 1-phenylethanol, and that suggests Wacker-type alkene oxidation given ketone did not occur. Co-oxidation of 1-phenylethanol and styrene also proceeded on the mixture of the non-Au-immobilized support Ti(1)-SBA^SH(0.5) and the non-titanium (IV)-loaded catalyst Au/Ti(0)-SBA^SH(0.5), but the yields of the oxidized products were decreased compared to those on Au/Ti(1)-SBA^SH(0.5). This result suggests that synergistic catalysis appears efficiently on the close arrangement of Au and Ti(IV). A trend of the yields of acetophenone which was derived from 1-phenylethanol was similar to that of the yields of styrene oxidized products.

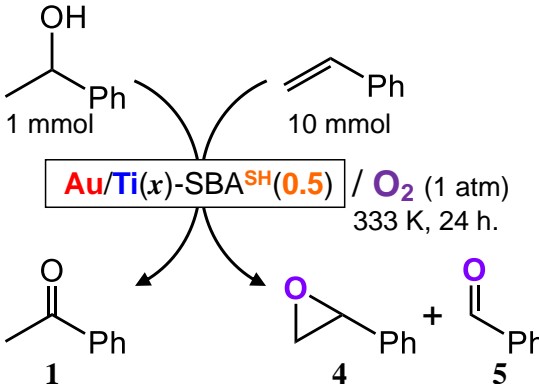

**Scheme 3.** Aerobic co-oxidation of 1-phenylethanol and styrene catalyzed by Au/Ti($x$)-SBA^SH(0.5).

**Table 4.** Conversion of 1-phenylethanol mediated by the calcined catalysts under $O_2$.

| Catalyst | Amount of Products/µmol | | | TON of Au for 1 | TON for 4 + 5 (Styrene Oxidation) | |
|---|---|---|---|---|---|---|
| | **1** | **4** | **5** | | **Based on Ti** | **Based on Au** |
| Au/Ti(0)-SBA^SH(0.5) | 51.2 | 1.2 | 17.3 | 36.6 | — | 13.2 |
| Au/Ti(0.5)-SBA^SH(0.5) | 106.1 | 0.1 | 30.0 | 78.6 | 9.1 | 22.3 |
| Au/Ti(1)-SBA^SH(0.5) | 58.3 | 2.5 | 55.4 | 38.9 | 7.6 | 38.6 |
| Au/Ti(2)-SBA^SH(0.5) | 96.9 | 1.3 | 83.2 | 66.8 | 5.5 | 58.3 |
| Au/Ti(1)-SBA^SH(0.5) [1] | 0 | 4.8 | 9.1 | — | 1.8 | 9.3 |
| Ti(1)-SBA^SH(0.5) + Au/Ti(0)-SBA^SH(0.5) [2] | 39.2 | 0.4 | 30.2 | 28.0 | 4.0 | 21.9 |

[1] Reaction without 1-phenylethanol; [2] Mixture of 50 mg of non-Au-supported Ti(1)-SBA^SH(0.5) and 50 mg of Au-immobilized non-Ti-involving support was used.

We examined the detection of the generated $H_2O_2$ in the absence of styrene. In the suspension including the catalysts, the amounts of detected $H_2O_2$ were consistent with the styrene oxidizing activity correlated with the loading amount of Ti(IV) (Figure 9a). In the liquid phase, however, no or small amounts of $H_2O_2$ were detected. Therefore, the formed $H_2O_2$ was absorbed on the surface of the catalysts. The detection of $H_2O_2$ even on the non-Ti(IV)-loaded catalyst suggests that the absorption on the silicate surface occurs.

In addition, the detection of the higher amount of $H_2O_2$ on the Ti-loaded catalysts can be explained why the complexation of $H_2O_2$ with the surface titanium occurs. The $H_2O_2$ trapping on the titanium(IV) sites might affect the alcohol oxidation activity. The order of the TON of Au on 1-phenylethanol oxidation was inverse of the loadings of titanium(IV); i.e., the non-titanium(IV)-functionalized catalyst Au/Ti(0)-SBA$^{SH}$(0.5) was the most active (Figure 9b).

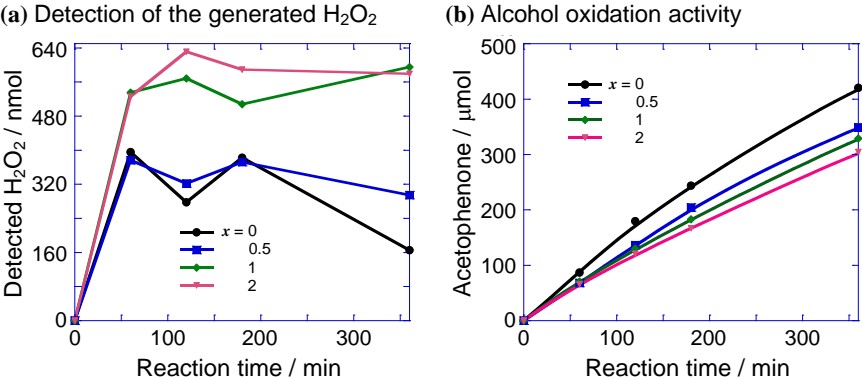

**Figure 9.** Effects of titanium sites on Au/Ti($x$)-SBA$^{SH}$(0.5): (**a**) Detection of $H_2O_2$ generated during the oxidation of 1-phenylethanol, (**b**) time course of 1-phenylethanol oxidation with $O_2$.

## 3. Discussion

In this work, we developed bi-functionalized SBA-15 type mesoporous silica by using a one-pot synthetic method successfully. The merit of the one-pot synthesis is the highly dispersed arrangement of the introduced functional groups [32]. In our developed supports Ti($x$)-SBA$^{SH}$($y$), the alkylthiol groups would be located on the wall of mesopores because the hydrophobic part of the alkylthiol precursor interacted with the polymer micelle templates during the formation of the silicate wall. The titanium(IV) ions were incorporated into the SiO$_2$ framework of SBA-15 and that resulted in the lower limit of the loading amounts of titanium(IV) (i.e., $x \leq 2$) because titanium(VI) favors octahedral rather tetrahedral coordination geometry. The catalytically active titanium(IV) species might locate at the surface of the silicate wall, and such titanium(IV) species were part of the whole titanium(IV) on Ti($x$)-SBA$^{SH}$($y$).

The correlation between the loading amounts of the alkylthiol groups and the immobilized amounts and the size of the particles of Au demonstrates that the alkylthiol groups worked as the initial binding site of ionic Au species and the template of the formed Au nano-particles. Noteworthy, the negatively- and positively-charged Au species formed on Au/Ti(0.5)-SBA$^{SH}$(0.5) and Au/Ti(0.5)-SBA$^{SH}$(8), respectively, as shown in Figure 5. Therefore, alkylthiol tunes the electronic property of the immobilized Au species.

The catalytic activity of the non-calcined catalysts toward the aerobic 2-phenylethanol oxidation depended on the loadings of alkylthiol (i.e., $y$ of Au/Ti(0.5)-SBA$^{SH}$($y$)). The most active catalyst was Au/Ti(0.5)-SBA$^{SH}$(0.5), of which the Au particles had 2–5 nm diameters and were charged negatively. Tsukuda and coworkers reported that the electron donation from Poly($N$-vinyl-2-pyrrolidone) to the Au cluster resulted in the negatively-charged Au species [33]. Ebitani and coworkers also reported the formation of negatively-charged Au species in the Au–Pd clusters [34]. The Au nano-particles on Au/Ti(0.5)-SBA$^{SH}$(0.5) were donated the electron from the thiolate donors. Those negatively-charged Au species activate the absorbed $O_2$ via the back-donation from Au 5d to π* orbital of $O_2$ giving superoxide ($O_2^-$) or peroxide ($O_2^{2-}$) like species bound to Au [35]. Therefore, the negatively-charged Au nano-particles formed on the support with $y$ = 0.5 was appropriate as the aerobic oxidation catalyst. On the higher alkylthiol-loaded supports with $0.5 < y \leq 8$, the formed Au species were rather positively charged and covered by alkylthiol. These electronic and structural properties might be a reason for the lower catalytic activity. The partial removal

of alkylthiol by calcination improved the activity of the catalysts with $0.5 < y \leq 8$. However, the activity of the calcined catalysts could not exceed that of Au/Ti(0.5)-SBA$^{SH}$(0.5).

The aerobic co-oxidation of 1-phenylethanol and styrene was efficiently promoted by the titanium(IV)-functionalized catalysts, whereas the activity of the non-titanium(IV) loaded catalyst was low. The styrene oxidation activity depended on the loading amounts of titanium(IV). Based on the assumption that the styrene oxidation proceeds only on the titanium site, the catalytic efficiencies of titanium(IV) estimated from the oxidized styrene products per the loaded titanium ion (i.e., TON of Ti(IV)) were inverse to the loading amounts $x$. As reported previously, higher loading of titanium led to the destruction of the ordered mesoporous structure of the titanosilicate due to the formation of titanium(IV) oxide (TiO$_x$) domain. In our catalysts, the higher loading of titanium(IV) might increase the inactive TiOx sites.

The results of the H$_2$O$_2$ detection experiments suggest the catalytic oxidation mechanism and the roles of the components of the present catalysts. Hydrogen peroxide is generated by the oxidation of alcohol by the gold catalyst. The resulting H$_2$O$_2$ is not only absorbed on the surface of the silica support but is also trapped by the titanium on the surface of the support. As a result of the trapping of H$_2$O$_2$ by titanium, the oxidation activity is reduced when alcohol is the only substrate. The highest 1-phenylethanol oxidizing activity on Au/Ti(0)-SBA$^{SH}$(0.5) may imply that the formed H$_2$O$_2$ is also utilized as the oxidant for alcohol oxidation. In the simultaneous oxidation of alcohols and alkenes, however, alkene oxidation activity is enhanced in the presence of titanium. Therefore, we can conclude that the titanium moiety contributes to the capture and activation of H$_2$O$_2$. In our catalysts, titanium might exist both on the surface and in the framework of the silicate wall. The titanium-containing supports absorbed a higher amount of H$_2$O$_2$ and that is evidence of the presence of titanium on the surface of the silica wall. However, only some titanium(IV) might be present on the surface. In order to improve the catalytic performance, the amount of the surface titanium species must be increased.

Finally, we will comment on the stability of the present catalyst. Checking the reusability of Au/Ti(0.5)-SBA$^{SH}$(0.5) on the aerobic oxidation of 1-phenylethanol revealed that the activity of the recovered catalyst was reduced (Figure S6 in Supplementary Materials). Measurement of the amount of immobilized gold onto the catalyst after the third use showed that approximately 8.2% of the initially loaded gold had been lost. The loss of gold may have been caused by a change in the Au-binding thiolate anion during the liquid phase oxidation reaction with a protic substrate, but the details are unknown. We are planning to investigate the catalytic activity of the gas-phase reaction in the future.

## 4. Materials and Methods
### 4.1. General

Nitrogen sorption/desorption studies were performed at liquid nitrogen temperature (77K) using a TriStar 3000 (Micromeritics Instrument Corporation, Norcross, GA, USA). Before the adsorption experiments, the samples were outgassed under reduced pressure for 3 h at 333 K. The X-ray diffraction data of the powder sample of the supports were collected a RINT-Ultima III (Rigaku, Tokyo, Japan). Inductively coupled plasma mass spectrometry (ICP-MS) was performed on a 7700 Series ICP-MS (Agilent Technologies, Santa Clara, CA, USA). Transmission electron microscopic (TEM) images were observed on a JEM-2010 (JEOL, Tokyo Japan) with an acceleration voltage of 200 kV and LaB$_6$ cathode. Scanned transmission electron microscopic (STEM) images were observed on a JEM-2100F (JEOL, Tokyo, Japan). Samples were prepared by suspending the catalyst powder ultrasonically in methanol and depositing a drop of the suspension on a standard copper grid covered with carbon monolayer film. EXAFS data were collected on BL01B1 at Spring-8 (Hyogo, Japan). X-ray photoelectron spectroscopy was measured on a JPS-9010 (JEOL, Tokyo, Japan) with a MgK$\alpha$ X-ray source (10 kV, 10 mA) for the analysis of the chemical states of the catalysts. The catalyst was pressed into a 20 mm diameter disk. Then, the disk was mounted on the sample holder of the XPS preparation chamber. The observed binding energy was

calibrated by using an $O_{1s}$ transition peak. Thermogravimetric analysis was performed on a Thermo plus EVO (Rigaku, Tokyo Japan). NMR spectra were recorded on an ECX-600 (JEOL, Tokyo, Japan). UV-vis spectra were measured on a V650 (Jasco, Tokyo, Japan). GC analysis was performed on a GC2010 (Shimazu, Kyoto, Japan) with an Rtx-1701 column (length = 30 m, i.d. = 0.25 mm, and thickness = 0.25 μm, Restek, Bellefonte, PA, USA).

All commercial reagents and solvents were used without further purification.

### 4.2. Preparation of the Bi-Functionalized SBA-15 Type Supports Ti(x)-SBA$^{SH}$(y)

The bi-functionalized mesoporous silica supports, Ti($x$)-SBA$^{SH}$($y$), were prepared similarly for the previously reported alkylthiol-functionalized SBA-15 by us [36]. To control the loading amounts of titanium(IV) and alkylthiol, the condensation ratio of tetraethoxysilane (Si(OEt)$_4$; $100 - (x + y)$ mol%), Titanium(IV) tetraisopropoxide (Ti(OEt)$_4$; $x$ mol%), and mercaptopropyltrimethoxysilane (Si(C$_3$H$_6$SH)(OMe)$_3$; $y$ mol%) were defined as $x$ = 0, 0.5, 1, 2, or 3, respectively, and $y$ = 0, 0.25, 0.5, 0.75, 1, 4, 8, or 10, respectively.

In a flask, 4.1 g of the surfactant Pluronic P123 (triblock copolymer EO$_{20}$PO$_{70}$EO$_{20}$ where EO = poly(ethylene oxide) and PO = poly(propylene oxide)) was placed, then dissolved in 150 mL of aqueous HCl solution (pH 3.0). To the resulting solution, appropriate amounts of the precursors (summarized in Table 1) were added. The mixture was stirred at 323 K for 20 h and subsequently heated and leave to stand at 373 K for 24 h. Once cooled, the solid product was filtered and washed with water and ethanol. The P123 surfactant was removed by Soxhlet extraction with a mixture of 100 mL of water and 200 mL of ethanol over a 24 h period. The resulting white solid was dried under a vacuum.

Loading amounts of titanium(IV) and alkylthiol were determined by UV-vis and 1H NMR spectrometry, respectively. The sample solution was prepared as follows. 10 mg of Ti($x$)-SBA$^{SH}$($y$) was dissolved in 2 mL of D$_2$O solution containing small amount of NaOH with heating. The quantitative analysis of titanium(IV) was done by using di-antipyrylmethane as the indicator for colorimetric quantitative analysis. For the quantity of alkylthiol, the NMR sample was prepared as follows. To the alkaline D$_2$O solution, 10 mg (0.072 mmol) of $p$-nitrophenol was added as an internal standard. The loading amounts of the thiol groups were estimated by comparison of the integration values of 1H NMR signals of ethylene and phenyl groups.

### 4.3. Preparation of the Catalysts Au/Ti(x)-SBA$^{SH}$(y)

In a flask, 0.50 g of the support Ti($x$)-SBA$^{SH}$($y$) (except $y$ = 0) were suspended in 30 mL of EtOH. In another flask, 0.01 g (0.025 mmol) of NaAuCl$_4$·2H$_2$O was dissolved in 20 mL of EtOH. The resulting EtOH solution of AuCl$_4^-$ was added dropwise to the suspension of the support, and the mixture was stirred for 12 h to adsorb gold ions on the support surface. The resulting solid was collected by filtration and washed with 400 mL of EtOH. The Au ions-absorbed support was suspended in 50 mL of EtOH. In a glass vessel, 20 mg (0.50 mmol) of NaBH$_4$ was dissolved in 5 mL of a mixture of H$_2$O and EtOH (1:1, $v/v$). The resulting NaBH$_4$ solution was added to the EtOH suspension of the Au ions-absorbed support. Then colored powder catalyst was collected, washed with 200 mL of EtOH, and dried under a vacuum.

The non-alkylthiol-functionalized support Ti(0.5)-SBA$^{SH}$(0) could not absorb Au ions. Therefore, Au/Ti(0.5)-SBA$^{SH}$(0) was prepared by the deposition of the in situ generated Au particles. To 0.5 g of Ti(0.5)-SBA$^{SH}$(0) suspended in EtOH, 0.01 g (0.025 mmol) of NaAuCl$_4$·2H$_2$O was added, then reduced by NaBH$_4$. The resulting purple-colored solid was collected, washed with EtOH, and dried under a vacuum.

The immobilized amount of Au was quantified by ICP-MS. The catalyst was dissolved in aqua regia, and the resulting solution was diluted with H$_2$O to apply ICP-MS.

### 4.4. Catalytic Aerobic Oxidation of 1-Phenylethanol

In the reaction vessel, 50 mg of the catalyst and 13 mg (100 mmol) of naphthalene (as an internal standard for GC analysis) were placed under an O$_2$ atmosphere. Then, 4 mL

of toluene (solvent) was charged and warmed at 333 K. Finally, 122 mL of (1.0 mmol) of 1-phenylethanol (substrate) was injected.

The by-produced $H_2O_2$ was analyzed by the method reported by Takamura and co-workers [37].

### 4.5. Catalytic Co-Oxidation of 1-Phenylethanol and Styrene

The reaction procedures were essentially similar to those of the aerobic oxidation of 1-phenylethanol (see above), although this co-oxidation reaction was performed without solvent. In the reaction vessel, 50 mg of the catalyst and 13 mg (100 mmol) of naphthalene (as an internal standard for GC analysis) were placed under an $O_2$ atmosphere. Then, 1.1 mL (10 mmol) of styrene (substrate) was charged and warmed at 333 K. Finally, 122 mL of (1.0 mmol) of 1-phenylethanol (substrate) was injected.

## 5. Conclusions

The titanium(IV)- and alkylthiol-functionalized SBA-15 type mesoporous silicate supports, $Ti(x)$-SBA$^{SH}(y)$, were synthesized by the one-pot condensation of the alkoxide precursors in the presence of the polymer micelle templates. The molar ratio of the precursors $Si(OEt)_4$: $Ti(OiPr)_4$: $Si(C_3H_6SH)(OMe)_3$ were controlled as $100 - (x + y)$: $x$: $y$ where $0 \leq x \leq 2$ and $0 \leq y \leq 8$, respectively. The supports synthesized with such composition showed the desired SBA-15 type characteristics, whereas the higher loadings of the functionalities (such as $x = 3$ or $y = 10$) destroyed the ordered mesoporous framework.

The catalysts $Au/Ti(x)$-SBA$^{SH}(y)$ were prepared by the reaction of the synthesized supports with $Au^{III}Cl_4{}^-$ and following reduction with $NaBH_4$. The structure and electronic properties of the immobilized Au species depended on the loadings of alkylthiol. The lower alkylthiol loaded support with $y = 0.5$ gave the negatively charged nano particulate Au species, whereas the higher loaded one with $y = 8$ yielded the positively charged Au clusters with the sulfur ligands. The aerobic alcohol oxidation activity also depended on the loadings of alkylthiol. The most active catalyst in a series of $Au/Ti(0.5)$-SBA$^{SH}(y)$ was that with $y = 0.5$, of which the negatively charged moderate-sized Au nano-particles might yield the active superoxide or peroxide adduct species. Both of the non-thiol-functionalized catalyst $Au/Ti(0.5)$-SBA$^{SH}(0)$ and higher alkylthiol-loaded catalyst $Au/Ti(0.5)$-SBA$^{SH}(8)$ were almost inactive due to the inactivity of the larger Au particles and the highly thiolate-covered small Au species.

The co-oxidation of styrene and 1-phenylethanol was promoted by $Au/Ti(x)$-SBA$^{SH}(0.5)$. The yields of the oxygenated compounds derived from styrene were increased on the increasing of the titanium loadings (i.e., $x$), although the major product was not styrene oxide but benzaldehyde. Hydrogen peroxide is generated by the oxidation of alcohol by the gold catalyst. The titanium moiety contributes to the capture and activation of $H_2O_2$.

Reuse of $Au/Ti(0.5)$-SBA$^{SH}(0.5)$ on the aerobic oxidation of alcohol resulted in reducing the catalytic activity due to the leaching of Au. The loss of Au might have been caused by a change in the Au-binding thiolate anion. We are planning to investigate the catalytic activity of the gas-phase reaction in the future.

**Supplementary Materials:** The following supporting information can be downloaded at: https://www.mdpi.com/article/10.3390/catal13010035/s1, Figure S1: $N_2$ adsorption isotherms of $Ti(0.5)$-SBA$^{SH}(y)$; Figure S2: $N_2$ adsorption isotherms of $Ti(x)$-SBA$^{SH}(0.5)$; Figure S3: XRD patterns of $Ti(0.5)$-SBA$^{SH}(y)$; Figure S4: XRD patterns of $Ti(x)$-SBA$^{SH}(0.5)$; Figure S5: Distributions of the particle size of the immobilized Au on $Ti(0.5)$-SBA$^{SH}(y)$ with $y = 0.5$ and 1; Figure S6: Reuse test of $Au/Ti(0.5)$-SBA$^{SH}(0.5)$ on the aerobic oxidation of 1-phenylethanol.

**Author Contributions:** Conceptualization, T.H., T.N., J.N. and S.H.; investigation, T.H. and T.N.; data curation, T.H., T.N., J.N. and M.O.; writing—original draft preparation, T.H.; writing—review and editing, M.O. and S.H.; supervision, S.H. All authors have read and agreed to the published version of the manuscript.

**Funding:** This research was funded by CREST, JST (JPMJCR16P1) and Kanagawa University (ordinary budget for 411).

**Institutional Review Board Statement:** Not applicable.

**Data Availability Statement:** The data presented in this study are contained within this article and are supported by the data in the Supplementary Materials.

**Conflicts of Interest:** The authors declare no conflict of interest.

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
