# Peer review of "Oxidation Catalysis of Au Nano-Particles Immobilized on Titanium(IV)- and Alkylthiol-Functionalized SBA-15 Type Mesoporous Silicate Supports"

_catalysts, doi:10.3390/catal13010035_

Round 1
Reviewer 1 Report
In this manuscript, the authors synthesized a series of Au nano-particle catalysts immobilized on titanium(IV)- and alkylthiol-functionalized SBA-15 type support. The chemical state and structure of Au on supports were investigated by changing the loading Ti and alkylthiol species. The catalytic performance of the various catalysts was systematacially studied for the aerobic alcohol oxidation. This contribution is of interest, and well written, and I positively recommend the publication of this manuscript in Catalysts after some modification.
The detailed concerns are listed as following:
1. The structure of the synthetic support is important, the SBA-15 structure of support with various Ti/alkylthiol loading should be identified by XRD.
2. The state of Ti-species in the support should be characterization, because the Ti-species could be located the framework of SBA-15 or the surface of SBA-15.
3. The cycle stability of catalysts should be evaluated.
Reviewer 2 Report
Journal: Catalysts (ISSN 2073-4344)
Manuscript ID: catalysts-2099116
Title: Oxidation catalysis of Au nanoparticles immobilized on titanium(IV)- and alkylthiol-functionalized SBA-15 type mesoporous silicate supports
Summary:
This paper shows the development of Au NPs catalysts supported on Titanium(IV)- and Alkylthiol-Functionalized SBA-15 Type Mesoporous Silicate Supports. The catalysts have been tested for the aerobic oxidation of 1-phenylethanol and of 1-phenylethanol and styrene.
After a careful reading of the manuscript, my opinion is that an improvement is required. The paper could be thus published after some major revision.
As general comment, both abstract and conclusion should be rewritten in a more appropriate, precise and clear way, reporting in more detail the results obtained. Also the introduction is rather poor and it should be thorough.
In particular,
in the abstract:
1. Not clear what is the reaction with AuCl4-: in the presence of the support? How does the immobilization take place?
2. “The aerobic alcohol oxidation activity of the catalysts depended on the loading of alkylthiol as well as the structure of the Au nanoparticles” – It should be mentioned which structure has one activity or another. The sentence is to general.
in the introduction:
3. Add references about Au nanoparticles for oxidation – for example:
- Gold catalysts for the selective oxidation of biomass‐derived products, S Cattaneo, et al. - ChemCatChem, 2019
- Oxidation of 5-hydroxymethylfurfural on supported Ag, Au, Pd and bimetallic Pd-Au catalysts: Effect of the support, S. Carabineiro et al. , Catalysts, 2021
- Glycerol oxidation over supported gold catalysts: The combined effect of Au particle size and basicity of support, E Kolobova et al. Processes, 2020
- Carbon-supported Au nanoparticles: catalytic activity ruled out by carbon support, A Jouve et al., Topics in catal., 2018
4. Too general references – they should be cited and described better and more in detail.
5. “The Ti(IV) site can activate H2O2 to oxidize hydrocarbons” – explain how with references
(general comment: Introduction is rather poor and must be it needs to be thorough.)
in the Results:
6. “The pore diameter distributions shown in 80 Figure x” – which figure?
7. “2.1.2. Preparation and characterization of Au-immobilized catalysts” - shouldn't that be the experimental part?
8. Fig. 4 – I suggested to insert in the manuscript bigger images. (to be more easily read)
9. XPS data and images: not very clear what the spectrum showed: did the authors perform some deconvolutions on the Au HR peaks?
10. “The XPS data also indicated that the cationic Au species converted to the 162 metallic one by calcination (Figure 7(b)).” – where are reported the B.E. values and, eventually, Au at. %?
in the Discussion:
11. “Noteworthy, the negatively- and positively- charged Au species formed on Au/Ti(0.5)-SBASH(0.5) and Au/Ti(0.5)-SBASH(8), respectively, as shown in Figure X.” – still present “figure x” , must be corrected.
12. “The anionic Au clusters are proposed to activate O2 giving a superoxide-like oxidant [19-22].” – this point should be better discussed with references
13. “The results of the H2O2 detection experiments suggest the catalytic oxidation mechanism and the roles of the components of the present catalysts.” – should be discussed
about Conclusion:
Incomplete – must be rewritten.
Round 2
Reviewer 2 Report
Dear authors,
after your revision I think that the paper is sufficiently improved to be published in catalysts.
Best regards